# Detection of Age-Related Changes in Tendon Molecular Composition by Raman Spectroscopy—Potential for Rapid, Non-Invasive Assessment of Susceptibility to Injury

**DOI:** 10.3390/ijms21062150

**Published:** 2020-03-20

**Authors:** Nai-Hao Yin, Anthony W. Parker, Pavel Matousek, Helen L. Birch

**Affiliations:** 1Research Department of Orthopaedics and Musculoskeletal Science, University College London, Royal National Orthopaedic Hospital, Stanmore HA7 4LP, UK; rmhknyi@ucl.ac.uk; 2Central Laser Facility, Research Complex at Harwell, Science & Technology Facilities Council, Rutherford Appleton Laboratory, Didcot OX11 0QX, UK; tony.parker@stfc.ac.uk (A.W.P.); pavel.matousek@stfc.ac.uk (P.M.)

**Keywords:** tendinopathy, ageing, Raman spectroscopy, collagen, autofluorescence, glycation

## Abstract

The lack of clinical detection tools at the molecular level hinders our progression in preventing age-related tendon pathologies. Raman spectroscopy can rapidly and non-invasively detect tissue molecular compositions and has great potential for in vivo applications. In biological tissues, a highly fluorescent background masks the Raman spectral features and is usually removed during data processing, but including this background could help age differentiation since fluorescence level in tendons increases with age. Therefore, we conducted a stepwise analysis of fluorescence and Raman combined spectra for better understanding of the chemical differences between young and old tendons. Spectra were collected from random locations of vacuum-dried young and old equine tendon samples (superficial digital flexor tendon (SDFT) and deep digital flexor tendon (DDFT), total *n* = 15) under identical instrumental settings. The fluorescence-Raman spectra showed an increase in old tendons as expected. Normalising the fluorescence-Raman spectra further indicated a potential change in intra-tendinous fluorophores as tendon ages. After fluorescence removal, the pure Raman spectra demonstrated between-group differences in CH_2_ bending (1450 cm^−1^) and various ring-structure and carbohydrate-associated bands (1000–1100 cm^−1^), possibly relating to a decline in cellular numbers and an accumulation of advanced glycation end products in old tendons. These results demonstrated that Raman spectroscopy can successfully detect age-related tendon molecular differences.

## 1. Introduction

Ageing is an important risk factor for developing tendon pathologies, which pose a significant medical burden for healthcare around the world. As a dense connective tissue with low cell density, the extracellular matrix (ECM) within tendons dictates mechanical properties and function. In addition to the direct structural supportive role of the ECM, it also affects cell behaviour through mechanotransduction [1,2,3]. Collagen, a major component of tendon ECM, assembles in a supra-molecular hierarchical structure with the ability to fulfil its structural role [4]. The exceptionally low turnover rate of collagen [5,6] makes it readily subjected to age-related enzymatic and non-enzymatic post-translational modifications [7]. The most important non-enzymatic modification is the glycation process, which forms multiple crosslinks within and between collagen molecules, thereby affecting their mechanical and biological properties [8,9]. Glucosepane, the most abundant advanced glycation end-product (AGE) in collagen-rich tissues, positively correlates with tendon age and affects tendon mechanical behaviour at various levels [10], suggesting its potential role in age-related tendon functional decline. Ageing is also associated with fragmentation of collagen in skeletal tissues [7,11]. These dysfunctional fragmented collagen molecules are thought to be retained in tendon, contributing to inferior mechanical properties and increased injury risk. However, these early ECM changes are non-symptomatic and difficult to assess in vivo using traditional imaging modalities. Biochemical analyses are time-consuming and require vigorous sample treatments in vitro, thereby hindering our ability to detect and successfully prevent age-related tendon pathologies outside the laboratory.

Alternatively, Raman spectroscopy can provide a comprehensive profile of the molecular composition of tissue in a rapid, label-free, and non-destructive manner [12]. This technique identifies molecular composition by measuring the inelastic scattering of the incident laser light. During the photon–molecule interaction, the molecule acquires vibrational energy while the scattered photon changes energy and, as such, its frequency. The shift in frequency, usually reported in wavenumbers (cm^−1^), depends on the molecular structure but is independent of incident light wavelength and can be used to probe the material composition [13]. The complex and overall chemical composition within the aggregated spectrum acquired by Raman spectroscopy can be analysed by using multivariate analysis methods enabling sample differences to be established [12,14] as, for example, healthy versus non-healthy tissue [15,16] or, indeed as we report here, discern tissue changes brought about by ageing [17]. Traditional Raman spectroscopy, when used with a microscope, has a spatial resolution of a few microns, making it an ideal tool for detecting regional heterogeneity within biological tissues [14,15]. For example, Raman spectroscopy combined with principal component analysis (PCA) enables differentiation of rat peripheral nerves from adjacent muscle or adipose tissues and also the separation of healthy, healing, or fibrotic tissue borders after myocardial infarction (see review [12]). Additionally, recent advances in developing spatially offset Raman spectroscopy (SORS) provide a means to obtain Raman spectra beneath the surface of diffusely scattering samples and has shown great potential in non-invasive in vivo measurements of underlying tissue molecular compositions [18,19]. SORS therefore has the potential to be employed as a clinical tool for identifying age-related tendon changes by giving molecular specific information going beyond X-ray and ultrasound methods.

Collagen forms approximately 80% of the dry weight of tendon, and therefore, Raman features of tendons are dominated by collagen signals [20]. These signals include the amide bands and the collagen backbone structure region. The amide bands relate to the amide vibrational patterns that indicate the secondary and tertiary structure of proteins, and the collagen backbone region relates to the C–C stretching and the quantity of proline and hydroxyproline. Previous studies applying Raman spectroscopy on tendons have identified different collagen types, crosslink profiles, secondary structures, and tendon fibril orientation and successfully detected degenerative lesions and distinguished stages of the healing process (see review [20] and [21,22,23,24]). Specifically, two studies suggested that ratios in the amide I and amide III bands could be used to identify age-related differences in other collagen-rich tissues by measuring mature crosslinks and disordered secondary structure of collagen [24,25]. However, the usefulness of these ratios in tendon analysis is currently unknown. The equine superficial digital flexor tendon (SDFT) represents a highly relevant tendon model in understanding sports-related and age-related tendon pathology and, to the best of our knowledge, has not been previously studied using Raman spectroscopy.

We hypothesised that Raman spectroscopy can differentiate young and old tendon matrix molecular composition in equine tendons with minimal sample processing (vacuum-dried and homogenised only). We chose the equine SDFT and deep digital flexor tendon (DDFT) due to their known functional and compositional differences despite lying anatomically adjacent to each other [26,27,28]. Generally, in Raman spectroscopy, a highly fluorescent background, as frequently encountered for biological samples, is seen as interference and is usually removed at the first stage in the data analysis. Typically, in biological tissues, the fluorescence level can be several orders of magnitude higher than the underlying Raman signal [29]. Many studies have reported instrumental modifications or mathematical methods for reducing the fluorescence influence (see review [30]) in order to correctly analyse the Raman signal. However, it is also reported that the fluorescence level of tendons increases with age, using traditional biochemical assays [6,26]. With this in mind, we have conducted a step-by-step analysis including both fluorescence-Raman combined spectra and pure Raman spectra (fluorescence removed) to establish a method for better understanding the age-related tendon molecular differences. We also calculated the two reported age-related ratios to explore the usefulness of these ratios in tendon analysis.

## 2. Results

In total, 150 spectra (10 spectra from each tendon sample) were acquired under identical conditions (100% laser power, 5 s exposure time, 10 acquisitions, 50× objective). The charge-coupled device (CCD, the signal detector of the system) counts were higher in old than young tendon samples (Figure 1). This unprocessed raw data suggested fluorescence level increases as both tendons age, similar to previous reports [26]. Since the Raman scattering intensity is several orders lower than the fluorescence intensity, normalisation and fluorescence removal were required to study the detailed Raman spectral signatures.

The spectra were then min–max normalised to study the relative influence of fluorescence and Raman scattering (Figure 2). Visual inspection suggested that the young group in both tendons showed stronger Raman intensities at various peaks than the old group after normalising the intensity level. PCA was then performed to objectively identify group separation (Figure 3). For both tendons, young and old groups were separated along the maximal variance axis (PC1), and cluster vector plots from PC1 of both tendons demonstrated collagen-related Raman features at wavenumbers between 800–1000 cm^−1^ (usually assigned to the collagen backbone, proline and hydroxyproline), amide III (C—N stretch and N—H bending, 1200–1300 cm^−1^), CH_2_ bending (1450 cm^−1^), and amide I (C=O stretch, 1600–1700 cm^−1^) bands. The PC1 vector loading confirmed our visual inspection result.

The fluorescence signal was then subtracted from each raw spectrum (i.e., untreated/unnormalised CCD counts) using a 6th order polynomial fit by a custom-written MATLAB code (MATLAB, The Mathworks, Inc., Natick, MA, USA) [16]. Removing the fluorescence signal produced a flat baseline permitting better visual interpretation of the Raman spectrum. Each baseline-corrected spectrum was then normalised to the amide III band (maximal intensity between 1230 and 1250 cm^−1^) [17] for comparison between different age groups and different tendons (Figure 4). Both SDFT and DDFT data were pooled for further PCA identification of age-related Raman spectral differences (Figure 4 and Figure 5).

PCA analysis of pooled Raman spectra (Figure 5) indicated that young and old groups were separated along the PC3 axis, which explained 4.7% of the total variance. Unsurprisingly, the axis of maximal variance (PC1, 34.5%) separated the two different tendons, indicating compositional differences between the two. Inspecting the vector loadings of PC1 and PC3 revealed that age-related Raman spectral features were different from compositional-related features. PC3 identified two large peaks at wavenumbers between 1000–1100 cm^−1^ (mainly ring-associated and carbohydrate bands) and 1400–1500 cm^−1^ (CH_2_ bending), and these two regions showed a clear difference in the pooled spectra (Figure 4). On the contrary, PC1 loadings suggested that SDFT and DDFT compositional differences were likely associated with different collagen types, since peaks between 800 and 1050 cm^−1^ are commonly assigned to proline (853, 934 cm^−1^), hydroxyproline (872 cm^−1^), collagen backbone (C–C stretching, 904 cm^−1^), and phenylalanine (990, 1003 cm^−1^) [31].

Univariate analysis of two reported age-related ratios (amide III ratio—random coil:alpha-helix [25] and amide I ratio—pyridinoline:dihydroxylysinonorleucine crosslinks [24]) showed no significant differences between young (amide III: 0.93 ± 0.07, amide I: 0.79 ± 0.08) and old (amide III: 0.95 ± 0.07, amide I: 0.79 ± 0.07) groups.

## 3. Discussion

The results of this study support the hypothesis and demonstrate that Raman spectroscopy can differentiate minimally processed young and old equine tendon tissue. In addition, our data show a clear difference in autofluorescence between young and old tendon tissue. Interestingly our results also show that it is possible to separate two different tendon types based on the Raman spectral features alone. PCA, an objective multivariate analysis tool, could separate the young and old tendons both in the fluorescence-Raman combined spectra and the Raman spectra alone and identified relevant molecular information.

Whole tissue fluorescence level was significantly higher in old than young tendons, similar to previous reports on equine tendons [26], rat Achilles tendon [32] and human patellar tendon [33]. However, it is important to note that the excitation wavelength is different between our Raman spectroscopy (830 nm) and the literature (295 nm [33], 330 nm [32], 348 nm [26]), so a direct comparison of the results cannot be made as our longer wavelength (near-IR) excitation can be expected to excite different chromophores to those in the previous studies. Fluorescence level has been used previously as a convenient measure of non-enzymatic glycation as many AGEs are fluorescent [10,34]. As AGEs accumulate with increasing chronological age, fluorescence level has also been used as an indicator of tendon matrix age or rate of turnover [26]. Due to the difference in excitation wavelength, the measured fluorescence in our study could result from intra-tendinous fluorophores other than the known collagen related AGEs, such as that associated with elastin [35], lipids [36], or cells [37]. Interestingly, after normalising the fluorescence intensity, the spectral shapes were found to remain different between groups, indicating that the range or ratio of fluorophores differs in young and old tendon tissue, and that multiple different fluorescence emission spectra overlap each other. Our results suggest that including raw spectral analysis, which contains the fluorescence signal, in the Raman data interpretation may enhance the ability to distinguish between tendons of different ages. Indeed, excluding this signal may be one of the reasons why fewer differences in fluorescence-removed Raman spectra between old and young tendons were detected in the previous report [17], despite using identical instrumentation and a similar protocol. Fluorescence has been used as an indirect measure of AGEs in human skin in vivo, and a positive correlation with age demonstrated, interestingly, a negative correlation with physical training [38,39,40], suggesting that physical activity can modify the accumulation of age-related chemical modifications. Spatially offset Raman spectroscopy provides the opportunity to measure tendon fluorescence level and Raman spectra simultaneously in a non-invasive manner [18,19,41], and could potentially detect age-related changes, as observed here, in tendon and other tissues in vivo.

Our fluorescence-removed Raman spectra showed a lower intensity of CH_2_ bending band (1450 cm^−1^) in old tendons compared to young tendons. The bending mode of the methylene group has usually been assigned to lipids or regarded as an indication of general protein content [31]. In equine SDFT and DDFT, the lipid content is low, as identified by mass spectroscopy [42], and any lipid-related signals are likely to originate from the cell membrane and may therefore be an indirect indicator of cell numbers. Previous studies using conventional biochemical assays have shown conflicting results with regard to cell numbers and change with age in both SDFT and DDFT [11,26,27,43]. A recent report, specifically investigating the cell count in tendon fascicles and interfascicular matrix (IFM), identified a trend towards decreasing cell numbers in the IFM but no difference in fascicles [43]. Raman spectroscopy has a high spatial resolution and can focus on the fascicle or IFM levels, providing an excellent tool for tendon researchers to better elucidate the age-related cell number change in specific anatomical regions or compartments. In the present study, we chose 10 randomly selected points on homogenised tendon samples for spectral collection in order to better reflect the whole tissue. Our result should be interpreted as the averaged spectra of various tendon compartments without potential selection bias. However, this prevents us from further investigating the potential spectral differences among compartments, especially between fascicles and interfascicular matrix. Since it has been shown that fascicles and interfascicular matrix have different proteomic profiles and demonstrate different age-related changes [43], differences in the Raman spectra can be expected. Future studies could use Raman spectroscopy to interrogate cells directly [12,14,15] and perform label-free classification of different cell populations in fascicles and the IFM, overcoming some of the current technological difficulties in studying tendon sub-structure composition.

Besides the CH_2_ bending band, the two groups (young and old) were also different in the various peaks within wavenumbers 1000–1100 cm^−1^. The peaks in this region are assigned to phenylalanine (1002, 1030 cm^−1^) or carbohydrates (1043, 1064, 1078 cm^−1^) [31,44]. However, phenylalanine only accounts for approximately 1% of amino acid residues in both alpha 1 and alpha 2 polypeptide chains of type I collagen [45] and is unlikely to change with tendon ageing. Both phenylalanine and carbohydrates have a carbon ring structure; hence, we tentatively assign these peaks in tendon tissue to other molecular compounds with aromatic rings or having sugar residues. The glycosaminoglycan (carbohydrate) component of proteoglycans has been shown to decrease significantly with increasing age in a positional equine tendon, although the decrease was not significant in the equine SDFT [6]. Another possibility is the non-enzymatic-mediated crosslinks pentosidine and glucosepane that both have ring structures and accumulate in tendon with age advancement [6,10]. In support of this, previous studies on in vitro induced collagen glycation showed a similar increase in peaks of this wavenumber region [46,47], and previous reports from our laboratory confirmed the accumulation of pentosidine and glucosepane in both equine [6,27] and human tendons [10]. AGE crosslinks between collagen molecules have been suggested to affect the mechanical behaviour at various tendon hierarchical levels, such as modifying the physical and mechanical properties of collagen molecules [48], fibres [49], and fibrils [10]; reducing the inter-fibre and inter-fibril sliding [49]; and altering viscoelastic properties at the gross tendon level [49]. In addition to biomechanical properties, glycation can affect cell-matrix interactions and take part in cell signalling pathways, stimulating the production of reactive oxygen species (ROS) and triggering an inflammatory response (see review [8]) providing a possible alternative aetiology for age-related tendon degeneration.

We found no between-group differences in the peak ratios within the amide III or amide I bands, in contrast to previously reported age-related changes in these ratios in bovine type I [24] and rat type II [25] collagen. This could be due to different experimental species or different tissues (bones and eyes versus tendons) or the relatively low matrix turnover in tendon, especially the collagenous matrix [6], after maturation compared to other connective tissues. This preliminary study demonstrates that future research is warranted for understanding the usefulness of these amide band ratios in tendons.

Our study demonstrated that Raman spectral features differ between tendon types. Despite their close anatomical location, the SDFT and DDFT are different in their matrix composition [26], which presumably results from their different functional demands. The SDFT, subjected to high stress and strain, stores and releases elastic energy for an energy-saving role during locomotion while the less strained DDFT mainly helps maintaining the stability of distal joints [28]. The SDFT has more type III collagen, more elastin, and a higher proteoglycan content than the DDFT [26,27]. Unsurprisingly, this clear compositional difference was identified by Raman spectroscopy in various bands (800–1000 cm^−1^) associated with the collagen backbone, proline, and hydroxyproline content. The water content also differs significantly between the SDFT and DDFT [26]. The level of hydration greatly influences the Raman spectrum of collagen [50,51], and since we could not maintain an exact tissue hydration level during our measurement period (~10 min), we chose to use dried tendon tissue to remove spectral variation due to different water content of samples.

In this proof-of-concept study, we have demonstrated that the non-destructive technique of Raman spectroscopy is able to discern age of tissue. At this stage, we have not sought to correlate the observed Raman spectral differences with the specific molecular changes brought about by tendon ageing. Although tendon is predominately type I collagen, there are many other minor components that may change with age and may be responsible for the differences observed in this study in the Raman spectra, as discussed above. In future studies, it will be necessary to conduct a detailed biochemical analysis of multiple components on the same tissue as that used in the Raman spectroscopy measurements and to undertake a comprehensive comparison of data to assign potential molecular features to spectral changes. Following initial assignment of molecular changes to specific Raman spectral features, experiments will need to be conducted with techniques to perturb the levels of individual molecules to test the proposed relationships.

In summary, we have identified that PCA provides a powerful unsupervised multivariate analysis tool and has been able to separate tendon compositional (PC1 axis) and age-related matrix changes (PC3 axis). Our fluorescence measurements also support the finding that tendon collagenous matrix shows very slow turnover after skeletal maturation [5,6], and age-related decline in tendon function is likely related to post-translational modifications of collagen, especially the glycation process. Overall, we conclude from this study that Raman spectroscopy has the potential to provide a useful clinical diagnostic tool to detect age- and disease-related changes to tendon and other ECM-rich tissues. Further work will be required to allow interpretation of the spectra into specific molecular differences between tendons and changes with age.

## 4. Materials and Methods

### 4.1. Tendon Sample Collection and Preparation

Equine forelimbs (*n* = 15) were collected from horses euthanased for reasons other than tendon injury at a commercial equine abattoir, as a by-product of the agricultural industry. Specifically, the Animal (Scientific Procedures) Act 1986, Schedule 2, does not define collection from these sources as scientific procedures. Tendons were carefully inspected during dissection to ensure that no tendons showing visible signs of pre-existing injury or degenerative changes were included in the study. All SDFTs (*n* = 7, young:old = 3:4) and DDFTs (*n* = 8, young:old = 4:4) were harvested within 24 h and kept frozen at −80 °C. Tendons were assigned into young (3 to 7 years old) or old (20 to 24 years old) group according to the horse age. On the test day, tendons were defrosted at room temperature and a small cubic of tissue taken (approximately 1 cm^3^) from the core. In the SDFT, tissue was taken from the mid-metacarpal level approximately 5 to 7 cm proximal to the metacarpal joint level. In the DDFT, the section site was approximately 2 cm proximal to the metacarpal joint level due to the compositional difference between the merging accessory ligament and tendon at mid-metacarpal level (unpublished data). All the tendon samples were then lyophilised and crushed, as part of a standardised process for other ECM biochemical analyses. The dried, powdered tendon samples were then analysed using Raman spectroscopy.

### 4.2. Raman Spectroscopy

All samples were analysed using a Renishaw inVia Raman spectroscopy (Renishaw, Gloucestershire, UK) equipped with an 830 nm laser (300 mW at source). Ten Raman spectra from each sample were acquired (5 s exposure time, 10 accumulations) from randomly selected locations to reflect the heterogeneity of tendon composition. The spatial resolution was 2 × 2 µm using a 50× objective.

### 4.3. Data analysis

All spectra were truncated to wavenumber 800 to 1800 cm^−1^ since this region provides important information regarding tendon matrix composition [16,17]. The truncated spectra were first compared between groups for understanding the discrepancy of fluorescence–Raman combined signal. Then, spectra were min–max normalised and underwent PCA (by Origin 2019, OriginLab Corporation, MA, US) to detect age-related differences containing both fluorescence and Raman signals. Scatter plots of principal components obtained from PCA were used for objective identification. The fluorescence signal was then removed from the unnormalised raw spectrum by an in-house MATLAB code using 6th order polynomial, and the subsequent Raman spectra were normalised to the amide III peak intensity (between 1230 to 1250 cm^−1^) [17]. Raman spectra of both SDFT and DDFT were then pooled and underwent PCA to identify the segregation of young and old tendons. Vector loadings of interested PC axes were plotted to visualise the variance in wavenumbers.

Univariate analysis was used for between-group comparisons of two age-related ratios. The calculation was based on the intensities of selected wavenumbers: amide III 1st peak to 2nd peak (1240 cm^−1^/1270 cm^−1^) [25], and the amide I pyridinoline to dihydroxylysinonorleucine ratio (1660 cm^−1^/1690 cm^−1^) [24].

## 5. Conclusions

We have demonstrated that Raman spectroscopy is able to detect age-related tendon molecular changes and to differentiate between tendon types. In addition, we further identify how Raman spectroscopy as a research tool, has the potential to be used in a clinic setting to monitor tendon health, detect premature/accelerated ageing, diagnose tendon disease, and track response to treatments for tendinopathies.

## Figures and Tables

**Figure 1 ijms-21-02150-f001:**
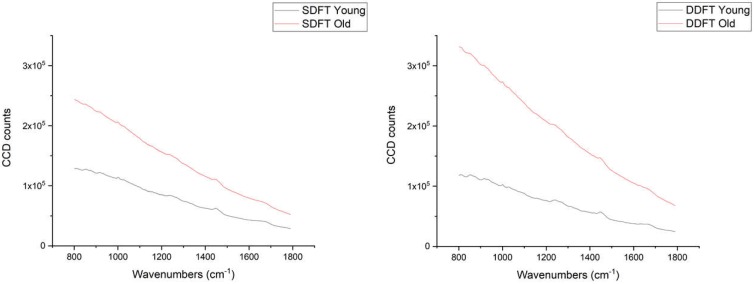
Averaged charge-coupled device (CCD) counts of young (40 spectra) and old (30 spectra) superficial digital flexor tendons (SDFTs, **Left**) and deep digital flexor tendons (DDFTs, **Right**) (40 spectra in both groups).

**Figure 2 ijms-21-02150-f002:**
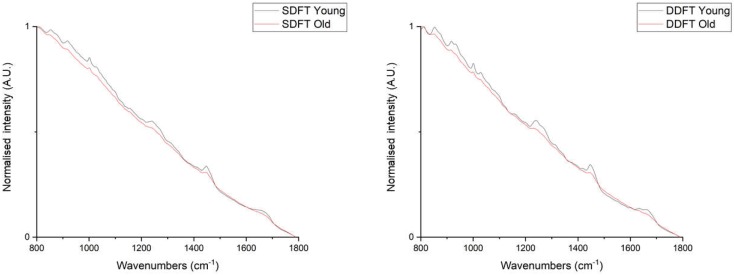
Averaged spectra after min–max normalised charge-coupled device (CCD) counts of young and old group in SDFTs (**Left**, young:old = 40:30 spectra) and DDFTs (**Right**, 40 spectra in both groups).

**Figure 3 ijms-21-02150-f003:**
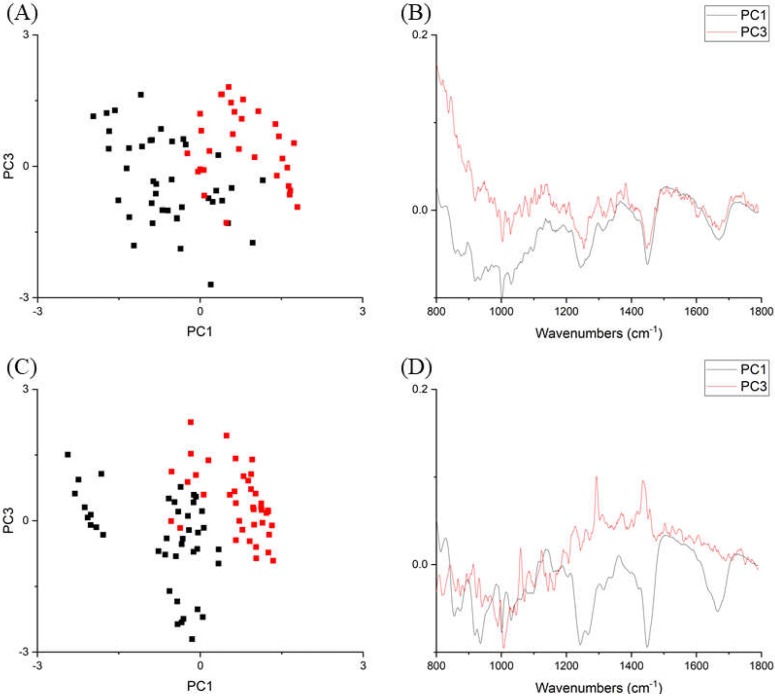
Principal component analysis on min–max normalised spectra. (**A**) SDFT, scatter plot of PC1 and PC3 axis. Red squares: young tendons; black squares: old tendons. (**B**) SDFT, PC1 (black line) and PC3 (red line) vector loading plot. (**C**) DDFT, scatter plot of PC1 and PC3 axis. Red squares: young tendons; black squares: old tendons. (**D**) DDFT, PC1 (black line) and PC3 (red line) vector loading plot.

**Figure 4 ijms-21-02150-f004:**
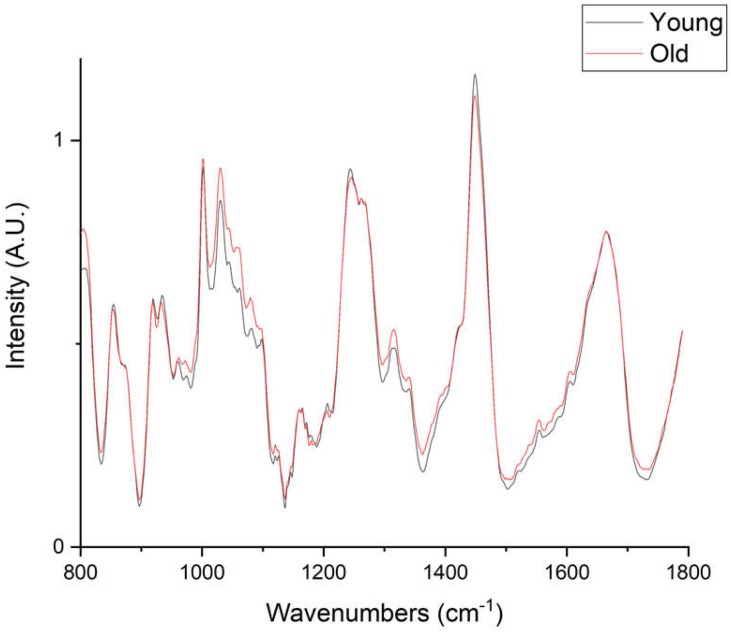
Averaged pooled Raman spectra of young (70 spectra) and old (80 spectra) tendons.

**Figure 5 ijms-21-02150-f005:**
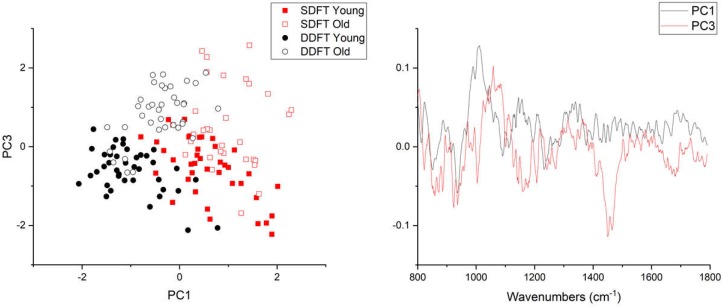
Principal component analysis of pooled Raman spectra. **Left**, scatter plot of PC1 and PC3 axis. PC1 separates tendon types and PC3 separates age groups. **Right**, PC1 and PC3 vector loadings.

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
