# Peer review of "Detection of Age-Related Changes in Tendon Molecular Composition by Raman Spectroscopy—Potential for Rapid, Non-Invasive Assessment of Susceptibility to Injury"

_ijms, 2020, doi:10.3390/ijms21062150_

Round 1
Reviewer 1 Report
Nai-Hao Yin, Anthony W. Parker, Pavel Matousek and Helen L. Birch reported a manuscript entitled, “Detection of age-related changes in tendon molecular composition by Raman spectroscopy – potential for rapid, non-invasive assessment of susceptibility to injury” to International Journal of Molecular Sciences.
The author sought to elucidate the age-related tendon pathologies by using Raman spectroscopy, which can reveal a non-invasively detect tissue molecular compositions and can have great potential for in vivo applications. The authors conducted a stepwise analysis of fluorescence and Raman combined spectra for better understanding the chemical differences between young and old tendons. Randomly obtained vacuum-dried young and old equine tendon samples (SDFT and DDFT, total n=15) were used and the fluorescence-Raman spectra showed an increase in old tendons as expected and normalizing the fluorescence-Raman spectra further indicated a potential change in intra-tendinous fluorophores as tendon ages. After fluorescence removal, the pure Raman spectra demonstrated between-group differences among CH2 bending (1450 cm-1) and various ring-structure and carbohydrates associated bands (1000-1100 cm-1), possibly relating to a decline in cellular numbers and an accumulation of advanced glycation end products in old tendons. The authors conclude that these results demonstrated that Raman spectroscopy can successfully identify age-related tendon molecular differences.
It in indeed very Interesting, intriguing and very attractive in clinical science of age-related tendon changes via non-invasive method and thus can eb repeatable and practical.
The data presented here are almost all understandable and scientifically sound, but thee relevance with the tissue, cellular and molecular changes to each Raman spectroscopy is lacking and highly recommended to include such data in the manuscript.
Minor points
In Materials and Methods, “Tendon sample collection and preparation”, The line “Equine forelimbs (n=15) were collected at a local abattoir where horses were euthanized for reasons other than tendon injury. “ should be further clarified with no affecting diseases or influential factors to the tendons tested.
Author Response
Response to Reviewer's Comments
We wish to thank the reviewer for their time in considering our manuscript and the very positive and encouraging comments. Please find below a detailed response to the comments and suggestions raised. We have made modifications to the manuscript to address the comments and these are highlighted in yellow. In addition, we have highlighted in green, text already in the manuscript relating to the comments made.
- English language and style
(x) Moderate English changes required
We have made modifications to the English language and grammar throughout the manuscript to improve the readability and the manuscript has been proof read by native English speakers.
- It in indeed very Interesting, intriguing and very attractive in clinical science of age-related tendon changes via non-invasive method and thus can eb repeatable and practical. The data presented here are almost all understandable and scientifically sound, but thee relevance with the tissue, cellular and molecular changes to each Raman spectroscopy is lacking and highly recommended to include such data in the manuscript.
We agree with the reviewer that the findings are interesting and very attractive for clinical translation and that the relevance of the findings to the molecular and cellular changes in the tendon tissue is important. We have added information in the discussion section to further explain the relationship between Raman features and molecular changes.
- Line number: 237 to 246 - The following text has been added, “previous reports from our laboratory confirmed the accumulation of pentosidine and glucosepane in both equine [6, 27] and human tendons [10]. AGE crosslinks between collagen molecules have been suggested to affect the mechanical behaviour at various tendon hierarchical levels, such as modifying the physical and mechanical properties of collagen molecules [52], fibres [53], and fibrils [10]; reducing the inter-fibre and inter-fibril sliding [53]; and altering viscoelastic properties at the gross tendon level [53]. In addition to biomechanical properties, glycation can affect cell-matrix interactions and take part in cell signalling pathways, stimulating the production of reactive oxygen species (ROS) and triggering an inflammatory response (see review [8]) providing a possible alternative aetiology for age-related tendon degeneration.”
- Line number: 250 to 251 - The following text has been added, “especially the collagenous matrix [6], after maturation compared to other connective tissues.”
To fully understand the complex relationship between spectral features and tendon composition, further experimental work is needed beyond the scope of this study and we have added two sentences to reiterate the need for further studies.
- Line number: 221 to 224 - The following sentence has been added, “Future studies could use Raman spectroscopy to interrogate cells directly [15, 50, 51] and perform label-free classification of different cell populations in fascicles and the IFM, overcoming some of the current technological difficulties in studying tendon sub-structure composition.”
- Line number: 273 to 274 - The following sentence has been added, “Further work will be required to allow interpretation of the spectra into specific molecular differences between tendons and changes with age.”
- In Materials and Methods, “Tendon sample collection and preparation”, The line “Equine forelimbs (n=15) were collected at a local abattoir where horses were euthanized for reasons other than tendon injury. “ should be further clarified with no affecting diseases or influential factors to the tendons tested.
This has been further explained by adding the sentence: “Tendons were carefully inspected during dissection to ensure that no tendons showing visible signs of pre-existing injury or degenerative changes were included in the study”. (line number 278-279)
Reviewer 2 Report
I have read the manuscript ID: ijms-733616 entitled „Detection of age-related changes in tendon molecular composition by Raman spectroscopy – potential for rapid, non-invasive assessment of susceptibility to injury“ with great interest. The draft fits in a perfect way into section to which the authors have submitted it: Molecular Pathology, Diagnostics, and Therapeutics. In order to further improve the manuscript I recommend that the authors have a close look into these two articles and extract the key-messages:
Label-free Molecular Imaging and Analysis by Raman Spectroscopy
Yasuaki Kumamoto, Yoshinori Harada, Tetsuro Takamatsu and Hideo Tanaka
Acta Histochem. Cytochem. 51 (3): 101–110, 2018 doi: 10.1267/ahc.18019
Raman Spectroscopy: Guiding Light for the Extracellular Matrix
Mads S. Bergholt, Andrea Serio and Michael B. Albro
Front, Bioeng. Biotechnol. 2019 Nov 1;7:303. doi: 10.3389/fbioe.2019.00303.
eCollection 2019.
Author Response
Response to Reviewer's Comments
We wish to thank the reviewer for their time in considering our manuscript and the very positive and encouraging comments. Please find below a detailed response to the comments and suggestions raised. We have made modifications to the manuscript to address the comments and these are highlighted in yellow. In addition, we have highlighted in green, text already in the manuscript relating to the comments made.
In order to further improve the manuscript I recommend that the authors have a close look into these two articles and extract the key-messages:
Label-free Molecular Imaging and Analysis by Raman Spectroscopy
Yasuaki Kumamoto, Yoshinori Harada, Tetsuro Takamatsu and Hideo Tanaka
Acta Histochem. Cytochem. 51 (3): 101–110, 2018 doi: 10.1267/ahc.18019
Raman Spectroscopy: Guiding Light for the Extracellular Matrix
Mads S. Bergholt, Andrea Serio and Michael B. Albro
Front, Bioeng. Biotechnol. 2019 Nov 1;7:303. doi: 10.3389/fbioe.2019.00303.eCollection 2019.
Thank you for highlighting these two very informative review papers. We have now cited these reviews (ref. 50 and 51) in our introduction and discussion. In addition, we have added text to draw out the key-messages:
- Line number: 65 to 68 - The following sentence has been added, “Raman spectroscopy combined with principal component analysis (PCA) enables differentiation of rat peripheral nerves from adjacent muscle or adipose tissues and also the separation of healthy, healing, or fibrotic tissue borders after myocardial infarction (see review [50])”
- Line number: 221 to 224 - The following sentence has been added, “Future studies could use Raman spectroscopy to interrogate cells directly [15, 50, 51] and perform label-free classification of different cell populations in fascicles and the IFM, overcoming some of the current technological difficulties in studying tendon sub-structure composition.”
Round 2
Reviewer 1 Report
Nai-Hao Yin, Anthony W. Parker, Pavel Matousek and Helen L. Birch reported a manuscript entitled, “Detection of age-related changes in tendon molecular composition by Raman spectroscopy – potential for rapid, non-invasive assessment of susceptibility to injury” to International Journal of Molecular Sciences.
The age-related tendon pathologies by using Raman spectroscopy, which can reveal a non-invasively detect tissue molecular compositions and can have great potential for in vivo applications. A stepwise analysis of fluorescence and Raman combined spectra for better understanding the chemical differences between young and old tendons. Randomly obtained vacuum-dried young and old equine tendon samples (SDFT and DDFT, total n=15) were used and the fluorescence-Raman spectra showed an increase in old tendons as expected and normalizing the fluorescence-Raman spectra further indicated a potential change in intra-tendinous fluorophores as tendon ages.
The authors conclude that these results demonstrated that Raman spectroscopy can successfully identify age-related tendon molecular differences.
It is necessary to confirm such age-related differences in molecular expression, regulation and histology or histochemical expressions for comparison and relevance.
Author Response
We appreciate that we are not able to identify specific age related changes in the tendon tissue and we have re-worded (line 29 and 272) to avoid ambiguity about this. In our original manuscript, we have stated that the primary aim of the study is to detect and differentiate between young and old using Raman spectroscopy (line 88, 172, 201, 230, 273, and 317). We agree that data on age-related molecular changes would be a very valuable next step to this work and we plan to seek funding to undertake this work. To provide meaningful data to link specific molecular differences to specific changes in the Raman spectra is a significant amount of complex work which will need to be carried out over the next two to three years and is beyond the scope of our 5 day resubmission time. Nevertheless we believe that the findings presented in the manuscript represent an exciting step forward in tendon research, which will be of interest to the readership of IJMS.
Round 3
Reviewer 1 Report
Nai-Hao Yin, Anthony W. Parker, Pavel Matousek and Helen L. Birch reported a manuscript entitled, “Detection of age-related changes in tendon molecular composition by Raman spectroscopy – potential for rapid, non-invasive assessment of susceptibility to injury” to International Journal of Molecular Sciences.
The authors have to prove the correlation between Raman spectroscopic findings and age-related tendon molecular changes with reference to other means such as histology, histochemistry, antibody-specific molecular expressions, otherwise one cannot understand the validity of the Raman spectroscopic changes leading to aging of the tendon.
Author Response
We have replied to the editor's comments and added a paragraph (line 266-276) in the discussion clearly stating our limitations."In this proof of concept study we have demonstrated that the non-destructive technique of Raman spectroscopy is able to discern age of tissue. At this stage we have not sought to correlate the observed Raman spectral differences the with the specific molecular changes brought about by tendon ageing. Although tendon is predominately type I collagen, there are many other minor components that may change with age and may be responsible for the differences observed in this study in the Raman spectra, as discussed above. In future studies it will be necessary to conduct detailed biochemical analysis of multiple components on the same tissue as that used in the Raman spectroscopy measurements and to undertake a comprehensive comparison of data to assign potential molecular features to spectral changes. Following initial assignment of molecular changes to specific Raman spectral features, experiments will need to be conducted with techniques to perturb the levels of individual molecules to test the proposed relationships." Academic Editor Notes The reviewers have pointed out clear improvements needed in the manuscript and by large the authors have addressed them.
The reviewer 1 has raised very important question regarding inclusion of relevant data on aging-related tissue, cellular and molecular changes to the Raman spectroscopic. This is very relevant question. However, in experimental terms to perform such study on the same horses used in this Raman evaluation is not realistic due to the non-invasive nature of the Raman spectroscopy as well as perhaps the need of ethical permission to sacrifice the horses. The authors intend to do such combined analysis in future.
The authors have already included in their discussion information on studies using conventional assays to study tendon aging. This section can be further expanded with details as well as the authors should include, as in their answer to the reviewer 1, in the discussion a paragraph on the limitation of performing biochemical analyses on the same horses, so it is clear for the reader.